# MultiMat: Multimodal Program Synthesis for Procedural Materials using Large Multimodal Models

**Jonas Belouadi**
University of Mannheim, Germany
jonas.belouadi@uni-mannheim.de

**Tamy Boubekeur, Adrien Kaiser**
Adobe Research, France
{boubek,akaiser}@adobe.com

## Abstract

Material node graphs are programs that generate the 2D channels of procedural materials, including geometry such as roughness and displacement maps, and reflectance such as albedo and conductivity maps. They are essential in computer graphics for representing the appearance of virtual 3D objects parametrically and at arbitrary resolution. In particular, their directed acyclic graph structure and intermediate states enable a modular, interpretable workflow for interactive appearance modeling. However, creating such graphs remains challenging and typically requires professional training. While recent neural program synthesis approaches attempt to simplify this process, they solely represent graphs as *textual* programs, failing to capture the inherently visual-spatial nature of node graphs that makes them accessible to humans. To address this gap, we present MultiMat, a *multimodal* program synthesis framework that leverages large multimodal models to process both visual and textual graph representations for improved generation of procedural material graphs. We train our models on a new dataset of production-quality procedural materials and combine them with a constrained tree search inference algorithm that ensures static correctness while efficiently navigating the program space. Our experimental results show that our multimodal program synthesis method is more efficient in both unconditional and conditional graph synthesis with higher visual quality and fidelity than text-only baselines, establishing new state-of-the-art performance.

## 1 Introduction

Procedural materials have become increasingly important in modern 3D content creation, offering artists greater control and flexibility in designing surface appearances for digital assets. Unlike traditional image-based textures, which are constrained by fixed resolutions and limited editability, procedural material modeling tools like Adobe Substance Designer (Adobe, 2025c) or Blender (Blender, 2025) leverage node-based graphs to generate textures programmatically. This enables resolution-independent execution, high-level parametric control, and non-destructive editing workflows that have proven valuable in industries such as game development, film production, and VR/AR applications (Musgrave et al., 2002). More specifically, a procedural material is defined as a directed graph where nodes represent texture generators (e.g., noise functions, patterns) or filtering operations (e.g., blurs, color adjustments), and edges encode the flow of data between these operations, ultimately producing the texture maps required by physically-based rendering (PBR) models (Pharr et al., 2016) (cf. Figure 1). However, the complexity of crafting these procedural material graphs presents a substantial barrier to entry, creating a pressing need for automated and semi-automated approaches to support material artists at all levels of proficiency.

With recent advances in neural program synthesis (Huynh & Lin, 2025), procedural material synthesis has become increasingly feasible. MatFormer pioneered this direction with a multi-stage transformer-based model for unconditional generation with Adobe Substance Designer (Guerrero et al., 2022). Building on this foundation, Hu et al. (2023) extended the approach to support conditional synthesis, enabling applications such as inverse rendering (Patow & Pueyo, 2003), i.e., generating procedural materials that match the appearance of captured or rendered images. More recently, VLMaterial demonstrated that large language models (Zhao et al., 2025) can effectively perform end-to-end procedural material synthesis (Li et al., 2025a). However, these approaches share a fundamental

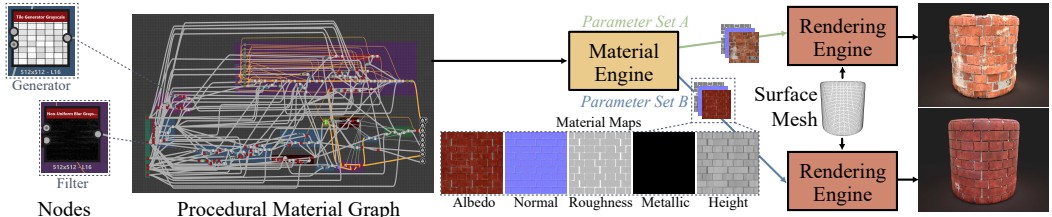

Figure 1: Procedural materials offer powerful control over the appearance of 3D objects through a few high-level parameters. Here, a production-grade example (left) with the images obtained using two distinct parameter sets A and B (right).

limitation: they generate node graphs as text-only programs without access to visual feedback during synthesis. This contrasts sharply with how human artists work, who create procedural materials by manipulating node graphs through an arguably more intuitive visual interface, as illustrated in Figure 1 (left). Without visual feedback, models must rely solely on textual representations to reason about complex spatial relationships and visual outcomes, a task that becomes increasingly difficult as material complexity grows. To address this limitation, we propose a novel *multimodal program synthesis* paradigm based on large multimodal models (Yin et al., 2024) that incorporates visual feedback throughout the generation process, more closely mirroring human creative workflows. We demonstrate that this approach, to which we refer as MULTIMAT, outperforms previous state-of-the-art methods (cf. §6). Our key contributions are as follows:

1. We introduce MULTIMAT, a novel procedural material synthesis approach that incorporates visualizations of intermediate graphs, including node states, into its context. This multimodal feedback loop improves material quality substantially compared to text-only baselines.

2. Investigating intermediate states enables real-time validation of each generated node. This allows us to develop a tree search algorithm that backtracks upon encountering invalid states, enabling more efficient inference than prior methods, which often produce invalid graphs.

3. We implement a transpiler that converts between Adobe Substance Designer formats and a compact representation suitable for language modeling while supporting the complete feature set. This enables training on larger datasets and the generation of more complex materials than previous approaches, which examined only limited subsets of Designer's capabilities.

## 2 RELATED WORK

**Large Language Models for Program Synthesis**  Our work builds upon recent advances in neural program synthesis (Parisotto et al., 2017; Devlin et al., 2017; Thakoor et al., 2018; Ye et al., 2021; Ellis et al., 2021). Traditional program synthesizers require formal specifications and employ search or logical derivation to produce programs that provably satisfy these specifications (Alur et al., 2013). Recently, large language models have demonstrated impressive capabilities in this domain (Huynh & Lin, 2025; Li et al., 2025b; Lozhkov et al., 2024; Li et al., 2023b; Rozière et al., 2023; Fried et al., 2023; Li et al., 2022; Chen et al., 2021). However, current research predominantly targets high-resource programming languages such as Python, Java, or JavaScript (Zan et al., 2023; Huynh & Lin, 2025). In contrast, our work synthesizes *graphics* programs, which pose unique challenges due to domain-specific requirements and considerable data scarcity, establishing it as a distinct research area.

**Graphics Program Synthesis**  Deep learning approaches have shown strong performance in synthesizing graphics programs that compile to visual outputs (Ellis et al., 2018; 2019; Ganin et al., 2018). This progress has been accelerated by the emergence of large multimodal models, particularly vision-language models that bridge visual and textual domains (Alayrac et al., 2022; Liu et al., 2023; Belouadi et al., 2024b; Kulits et al., 2024; Li & Ellis, 2024; Kapur et al., 2025; Lin et al., 2025; Xu et al., 2025). The field encompasses both controlled experimental settings using domain-specific languages (Ellis et al., 2018; Tian et al., 2019; Sharma et al., 2018; Cámara et al., 2023; Kulits et al., 2024; Kapur et al., 2025; Wen et al., 2025) and practical applications. Notable examples include

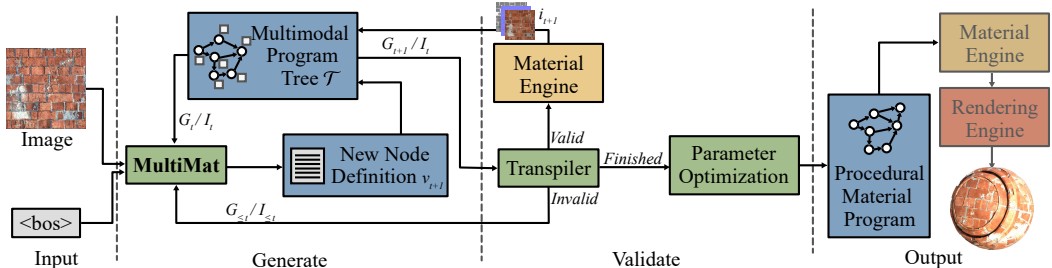

Figure 2: Architecture overview of MULTIMAT during inference. The system constructs a multimodal program tree $\mathcal{T}$ by iteratively generating node definitions. At each step $t$, the system derives a graph $G_t$ of valid nodes along with corresponding intermediate outputs $I_t$ by traversing $\mathcal{T}$, which may contain both valid and invalid nodes, to generate the next node $v_{t+1}$. When transpilation and execution succeed, the system advances with an updated graph $G_{t+1}$ and outputs $I_{t+1}$. If errors occur, it reverts to a previous state $(G_{\leq t}, I_{\leq t})$. The generation process initiates from either an input image or unconditionally using a beginning-of-sequence token (<bos>). Following optional parameter optimization (cf. §6.2), the final procedural material can be applied to any target geometry for rendering.

systems for generating scientific figures using TikZ (Belouadi et al., 2024a;b; 2025; Laurençon et al., 2024; Laurençon et al., 2024; Tong et al., 2024; Zhang et al., 2025) and automating data visualization (Mackinlay, 1986; Roth et al., 1994; Luo et al., 2021; Wu et al., 2024; Voigt et al., 2024). However, these approaches generate code designed for text-based editing and therefore do not face the unique circumstances of node graphs in procedural material synthesis that our work addresses.

**Procedural Material Synthesis** Procedural material modeling is one of the most challenging domains in graphics program synthesis. The combination of lengthy, complex material programs and severe data scarcity creates unique obstacles for learning-based approaches (Li et al., 2025a; 2024). Existing methods primarily focus on inverse procedural material modeling by synthesizing graphs that reproduce a given target appearance (Hu et al., 2023) or unconditional generation to create diverse, novel materials without specific targets (Guerrero et al., 2022). A related line of work optimizes parameters of existing material graphs to match image targets by transpiling them into differentiable programs (Shi et al., 2020; Hu et al., 2022; Li et al., 2023a). As discussed in §1, previous generative approaches are limited to text-only representations, a limitation we address in this work.

## 3 BACKGROUND ON PROCEDURAL MATERIALS

As indicated in §1, procedural materials are directed acyclic graphs $G$, executed by a material engine to produce raster images representing the physical properties of materials. These so-called material maps define surface characteristics, e.g., albedo, roughness, or normal (tangent space orientation), that enable photorealistic rendering when applied to 3D objects, with their appearance controlled through a small set of high-level parameters (cf. Figure 1). The internal structure of a material graph $G$ comprises nodes $\{v_1, v_2, \ldots, v_N\}$ connected by edges that define the flow of image data. Each node $v_i$ functions as either a generator that creates new image content or a filter that transforms existing images from upstream nodes. Common node operations include noise generation, blending, and mathematical transformations, which collectively produce intermediate image outputs $I = \{i_1, i_2, \ldots, i_N\}$. The behavior of each node is governed by parameters that may be discrete or continuous scalars or vectors, providing fine-grained control over the final material appearance.

Professional material authoring tools such as Blender and Adobe Substance Designer enable artists to construct and modify procedural material graphs through visual interfaces (cf. Figure 1). Users can interactively add or remove nodes and edges while adjusting node parameters to achieve desired visual effects. Among these tools, Adobe Substance Designer stands out for its particularly expressive node graph system, which MULTIMAT specifically targets. It offers advanced capabilities for creating complex material appearances through features like function graphs and pixel processors. Function graphs allow parameters to be controlled through custom operations on input values, while pixel

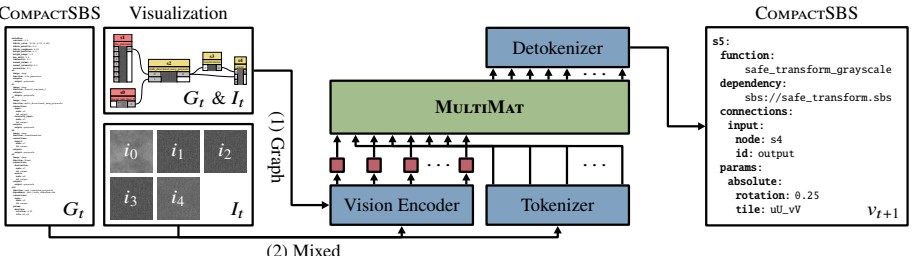

Figure 3: Visualization of the two conditioning approaches used by MULTIMAT for generating node definition $v_{t+1}$. In the graph-conditioned approach (1), MULTIMAT processes the graph $G_t$ as a visual representation similar to human perception. In the mixed-conditioned approach (2), MULTIMAT receives $G_t$ as a multimodal program where `` tokens are replaced with their corresponding vision encoder representations from $I_t$.

processors enable users to define specialized computational graphs that operate on individual pixels using sequences of atomic mathematical operations. These sophisticated capabilities make automated procedural material synthesis a particularly challenging problem in this domain.

## 4 THE MULTIMAT MODEL & ARCHITECTURE

Figure 2 illustrates our complete model pipeline. At its core, MULTIMAT is a vision-language model, trained for synthesizing procedural material graphs. It accepts images as input for inverse procedural material synthesis and supports unconditional generation. Unlike previous approaches, MULTIMAT generates nodes *topologically*, ensuring each node precedes all nodes it connects to. This enables an iterative generation process detailed below that can provide continuous visual feedback to the model, verify the validity of intermediate outputs, and recover from errors automatically in certain cases.

### 4.1 MULTIMODAL PROGRAM SYNTHESIS

Given a partially generated material graph $G_t = \{v_1, v_2, \ldots, v_t\}$ with nodes $v_i$ at generation step $t$, the topological ordering of nodes allows for visualizing intermediate node states, similar to visual editing environments that target humans. This enables an iterative generation loop where MULTIMAT generates one node definition—including node parameters and connections to previous nodes—at a time that is processed accordingly before the generation continues. After generating node $v_{t+1}$ in an intermediate text format (cf. §5), we combine it with the existing node definitions $\{v_1, \ldots, v_t\}$ and feed them to a transpiler, which compiles the intermediate representations back to a format the material engine understands. We then use the material engine to visualize the state of node $v_{t+1}$. Upon successful transpilation and execution, $v_{t+1}$ is appended to the graph $G_t$, resulting in $G_{t+1}$. This updated state, including the visualized intermediate outputs $I_{t+1}$, is fed back to the model to generate the subsequent node $v_{t+2}$ (cf. Figure 2). If execution or transpilation fails, we discard the current $v_{t+1}$ and resample, or backtrack further in case of repeated errors (cf. §4.2). We explore two complementary approaches for representing $G_t$ and $I_t$ as *multimodal programs* to the model, as visualized in Figure 3:

**Mixed Conditioning** Starting with a textual representation of $G_t$ (cf. §5), we enhance each node $v_i$ with an additional field containing its visualized intermediate state. This creates a multimodal program where the model processes textual tokens interleaved with image patch embeddings (cf. Figure 3). To manage the increased context size from image embeddings, we omit node parameters (which are implicitly encoded in the visualizations) but explicitly include node output type information (e.g., grayscale or color) that the model cannot infer from the visualization alone.

**Graph Conditioning** This approach more closely mirrors human visual experience by conditioning MULTIMAT solely on a visualization of the entire graph $G_t$ with embedded intermediate visual outputs $I_t$, as shown in Figure 3. The model generates subsequent node $v_{t+1}$ using only this complete visual context, without explicit access to underlying textual node definitions.

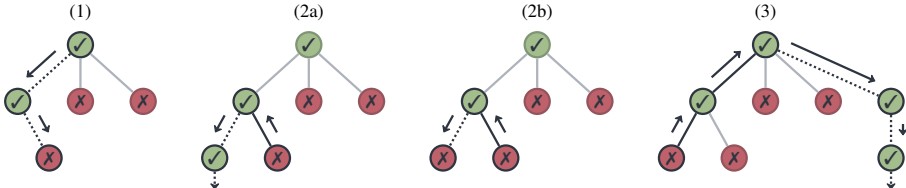

Figure 4: Visualization of our inference algorithm as a tree search. Tree nodes represent generated node definitions, and edges represent possible continuations. The algorithm proceeds as follows: generation continues until an invalid state (✗) is encountered (1), triggering backtracking to the previous node; from this point, if a valid node (✓) is generated, normal generation resumes (2a), but if invalid outputs persist (2b), the algorithm backtracks further until a valid path is found (3).

At their core, both approaches remain autoregressive language models, and MULTIMAT can be trained by minimizing a cross-entropy objective:

$$\mathcal{L} = -\sum_{t=1}^{T} \sum_{s=1}^{S} \log p(\, v_{t,s} \mid v_{t,<s}, G_t, I_t, x; \theta\,), \tag{1}$$

where $v_{t,s}$ is the expected token of $v_t$ in our intermediate text format at position $s$, $v_{t,<s}$ represents previous tokens, $x$ denotes the input conditions (which can be empty for unconditional generation), and $\theta$ represents the model parameters.

## 4.2 INCREMENTAL TREE SEARCH

Another advantage of topological node ordering is the ability to validate node definitions incrementally during generation. By invoking our transpiler and material engine at each step, we can detect syntactic and semantic errors immediately rather than waiting until the entire graph is complete. When an erroneous node definition is encountered, we execute an adaptive backtracking strategy: first discarding and resampling the problematic node, and if errors persist, inferring deeper structural issues by reversing further back in the generation sequence. Specifically, we discard the $2^{(i-1)}$ most recently generated nodes, where $i$ represents the current backtracking iteration. This approach effectively transforms our generation process into an *incremental tree search* on a tree $\mathcal{T}$ of valid and invalid nodes (cf. Figure 4), systematically exploring the solution space to discover valid programs. This incremental validation approach identifies invalid outputs much faster than previous approaches, which require sampling complete programs before validation can commence.

## 4.3 AUTOMATIC ERROR REPAIR

Through systematic analysis of failure cases, we identified recurring error patterns that could be repaired automatically: (1) removal of extraneous parameters that are specified for node types that do not support them, and (2) automatic insertion of conversion nodes to resolve type mismatches between connected nodes. For instance, when a color output is erroneously connected to a grayscale input, we automatically insert an appropriate grayscale conversion node. Conversely, when a grayscale output feeds into a color input, we insert a gradient map node to perform type conversion. These repair mechanisms increase the proportion of valid generations without requiring additional sampling steps.

## 5 DATASET

To support the training and evaluation of MULTIMAT, we collect procedural materials from Adobe's Substance 3D Assets Repository (Adobe, 2025a). Unlike previous work that either focuses on basic graphs utilizing only a subset of Substance Designer features (e.g., lacking complex nodes such as pixel processors or function graphs; Guerrero et al., 2022; Hu et al., 2023) or targets other tools with more limited capabilities (Li et al., 2025a), our approach supports the complete feature set. This comprehensive coverage enables us to collect over 6 000 unique materials, substantially more than existing datasets. Table 1 summarizes key characteristics of our dataset compared to prior work.

**Human-Readable Graph Representation** Substance Designer's native file format (SBS) has not been designed for human readability, containing verbose XML structures, embedded binary data, legacy metadata, and other implementation details, which makes direct language modeling impractical. To address this, we develop

| Models | Size | Max Nodes | Feature Set | Program |
|---|---|---|---|---|
| MatFormer | 2 820 | $\leq 400^{1}$ | Subset | Designer |
| Mat. (Cond) | 4 667 | $\leq 80^{1}$ | Subset | Designer |
| VLMaterial | 3 663 | 30 | Limited | Blender |
| MultiMat | 6 878 | 128 | Complete | Designer |

[1] Upper bound in complex filtering pipeline, actual could be less.

Table 1: Comparison of training data of MatFormer (Guerrero et al., 2022), conditional MatFormer (Hu et al., 2023), VLMaterial (Li et al., 2025a), and MultiMat (ours). We assembled the largest dataset with the most comprehensive set of features.

a bidirectional transpiler that converts between SBS and a compact, human-readable YAML-based representation with topological node order, which we call *CompactSBS*. Unlike previous approaches that support only partial feature sets (Guerrero et al., 2022; Hu et al., 2023), our transpiler preserves the complete functionality of Substance graphs with programs that are, on average, over 80% shorter. Models operate exclusively in CompactSBS, with outputs transpiled back to SBS for execution. We provide representative examples in Figure 3 and complete program listings in Appendix A.

**Graph Preprocessing** Our preprocessing pipeline standardizes graphs for the PBR workflow, focusing on five essential texture maps: base color, normal, roughness, metallic, and height. We trace backwards from these outputs to identify all contributing nodes, pruning unconnected components and other output maps. Graphs containing embedded bitmap graphics and SVGs are excluded to keep graphs fully procedural. We further filter out graphs exceeding 128 nodes and flatten hierarchical structures by inlining nested subgraphs and custom author dependencies into the main graph. Non-atomic nodes from the standard Substance Designer library remain as external references.

## 6 Experiments

We build MultiMat models upon the Qwen2.5$_{VL}$ (7B) vision-language model which leverages a late fusion approach to combine image and text tokens (Bai et al., 2025). We train and evaluate separate models for unconditional generation (cf. §6.1) and inverse procedural material synthesis (cf. §6.2). We also conduct a human evaluation (cf. §6.3). Across all model variants, we maintain a consistent maximum sequence length of 8 192 tokens. The training setup consists of 5 epochs using AdamW (Loshchilov & Hutter, 2019), a learning rate of 5e−5, and a batch size of 128. To ensure diversity in our generated outputs, we set the inference sampling parameters to a temperature of 0.8 and a top-p value of 0.95. We provide examples in Figure 6 and Appendix A. We ablate incremental tree search in §7.

### 6.1 Evaluation of Unconditional Generation

For unconditional generation, the mixed conditioning variant, *MultiMat (Mixed)*, embeds node previews at $140 \times 140$ resolution, resulting in 25 patch embeddings per image. For the graph conditioning variant, *MultiMat (Graph)*, graph visualizations can utilize up to 6 144 tokens, with larger images downscaled to accommodate this limit. We generate 100 outputs per model for evaluation.

**Baselines** For text-only procedural material synthesis, VLMaterial represents the current state-of-the-art approach. However, its Blender-specific training makes direct comparison with our method difficult. We therefore create VLMaterial (SBS) by retraining a VLMaterial-style model on our dataset for fair comparison. Unlike the objective in equation (1), VLMaterial is trained to generate complete graphs in a single pass. However, during inference, we can still validate nodes as they are generated and roll back upon detecting irreparable errors (cf. §4.2) or repair them after generation completes (cf. §4.3). This means the progression from VLMaterial (SBS) to MultiMat (Mixed) to MultiMat (Graph) represents a comparable, gradual shift from complete graph generation toward iterative node generation. Since VLMaterial (SBS) does not receive any images in the unconditional setting, we base it on the larger and more powerful text-only model Qwen3 (8B; Yang et al.,

| Models | DSIM↑ | CLIP↑ | STYLE↓ | KID↓ | ROUGE-L↓ | NER↓ |
|---|---|---|---|---|---|---|
| VLMATERIAL (SBS) | 31.344 | 65.678 | 3.211 | 14.976 | **1.621** | 16.933 |
| MULTIMAT (Mixed) | 34.922 | 66.737 | 3.199 | 3.675 | 2.194 | **12.388** |
| MULTIMAT (Graph) | **36.609** | **67.907** | **3.178** | **2.801** | 2.037 | 17.046 |
| VLMATERIAL⁺ (SBS) | 31.348 | 65.867 | 3.126 | 27.862 | | |
| MULTIMAT⁺ (Mixed) | 40.258 | 69.687 | 3.093 | 17.792 | | |
| MULTIMAT⁺ (Graph) | **40.367** | **70.114** | **3.046** | **14.886** | | |

Table 3: System-level scores × 100 for conditional (inverse) generation, without (top) and with (bottom) parameter optimization. Bold and underlined values indicate the best and second-best scores for each metric column, respectively. Arrows indicate metric directionality. ROUGE-L and NER scores remain unchanged by parameter optimization and are shown only once. MULTIMAT (Graph) and MULTIMAT⁺ (Graph) achieve the best overall performance.

| MULTIMAT⁺ (Mixed) | 83.3% | 16.7% | VLMATERIAL⁺ (SBS) |
| MULTIMAT⁺ (Graph) | 81.1% | 18.9% | VLMATERIAL⁺ (SBS) |
| MULTIMAT⁺ (Graph) | 58.3% | 41.7% | MULTIMAT⁺ (Mixed) |

Figure 5: Human preferences for model outputs as a diverging bar chart. MULTIMAT⁺ (Graph) is the most preferred model overall, while VLMATERIAL⁺ (SBS) is consistently the least preferred.

2025a), giving it a slight advantage over our models. While graphics program synthesis research typically also benchmarks against proprietary large language models such as GPT-4o (OpenAI et al., 2024) or CLAUDE 4 (Anthropic, 2025), which have demonstrated competitive performance in related domains (Belouadi et al., 2024a;b; 2025; Rodriguez et al., 2025), these models' unfamiliarity with COMPACTSBS and inability to produce valid SBS output preclude their inclusion as baselines.

**Metrics** Our multimodal task permits diverse evaluation schemes for automatic evaluation. To evaluate the *visual quality* of generated materials, we compute the Kernel Inception Distance (**KID**; Bińkowski et al., 2018), which compares the distribution of generated material maps with material maps from our dataset. To detect degenerate low KID scores due to *memorization* of training data (a legitimate concern given our relatively small dataset), we also calculate **ROUGE-L** scores (Lin, 2004) between the COMPACTSBS representation of our generated materials and the training set (with masked parameters). This metric computes the longest common subsequence and serves as an effective memorization indicator (Hans et al., 2024). Notably, we specifically require *consecutive* subsequences due to COMPACTSBS's limited syntactic diversity, which could otherwise produce misleading matches. To measure *efficiency*, we introduce the Node Error Ratio (**NER**), defined as the average ratio between discarded nodes and the total number of generated nodes.

**Results** Table 2 presents the system-level metric scores for our evaluation. MULTIMAT (Graph) leads in visual quality with the lowest KID score, outperforming MULTIMAT (Mixed) by over 4pp (percentage points) and VL-MATERIAL (SBS) by more than 11pp. This considerable gap in performance suggests that the better the visual representations are aligned with human creative workflows, the better the results— an intuitive but important finding. All

| Models | KID↓ | ROUGE-L↓ | NER↓ |
|---|---|---|---|
| VLMATERIAL (SBS) | 14.155 | 3.641 | 14.846 |
| MULTIMAT (Mixed) | 6.752 | 2.195 | **8.923** |
| MULTIMAT (Graph) | **2.365** | **1.915** | 15.024 |

Table 2: System-level scores × 100 for unconditional generation. Bold and underlined values indicate the best and second-best scores for each metric column, respectively. Arrows indicate metric directionality. MULTIMAT (Graph) achieves the best overall performance.

models exhibit minimal memorization, with ROUGE-L scores showing that no more than 4% of any generated sequence matches a contiguous segment from the training data. Nonetheless, both MULTIMAT variants demonstrate approximately 1.5pp lower copying rates compared to VLMATERIAL (SBS), suggesting slightly better generalization. Regarding efficiency, MULTIMAT (Mixed) excels with

the lowest NER, achieving a 6pp improvement over the other models. Both MULTIMAT (Graph) and VLMATERIAL (SBS) show comparable NER scores around 15%. For MULTIMAT (Graph), these errors are primarily due to OCR-like errors in reading node names and function types embedded as text in graph images. In contrast, we attribute the errors in VLMATERIAL (SBS) to fundamental difficulties in understanding graph structures. Despite these limitations, the error rates remain within acceptable bounds for practical applications, and MULTIMAT (Graph) emerges as the best overall model.

## 6.2 EVALUATION OF CONDITIONAL GENERATION

As in prior work (Hu et al., 2023; Li et al., 2025a), we train inverse MULTIMAT variants that learn to generate procedural materials from rendered images. These models follow the same training procedure as their unconditional counterparts, with one key modification: each training example is preceded by a $512 \times 512$ rendering of itself, which adds 324 additional image patches to the model context. During inference, the model takes an image as input and generates a corresponding procedural material. We reserve 100 examples from our data as held-out test data for evaluation.

**Baselines** Analogously to §6.1, we adapt VLMATERIAL for inverse rendering with SBS and use it as a baseline. Since an image input is now required for VLMATERIAL (SBS), we also base it on QWEN2.5$_{VL}$ (7B) instead of QWEN3 (8B) and train it using the same method as MULTIMAT.

**Parameter Optimization** To further refine generated materials, we apply gradient-based optimization using differentiable rendering. This approach has proven effective for optimal parameter estimation (Shi et al., 2020; Hu et al., 2022; Li et al., 2023a; Hu et al., 2023). We employ DiffMat (Shi et al., 2020; Li et al., 2023a), a widely adopted differentiable renderer for Designer materials, to optimize the generated graphs against the input images. Models using this refinement step are denoted as MULTIMAT$^+$ and VLMATERIAL$^+$, respectively.

**Metrics** In addition to the metrics from §6.1, we evaluate reconstruction quality by rendering the generated materials and comparing them to the input images using perceptual similarity metrics. Specifically, we measure cosine similarity between **CLIP** image embeddings (Radford et al., 2021; Hessel et al., 2021), compute STYLE Loss loss (**STYLE**; Gatys et al., 2016) as the L1 distance between Gram matrices of VGG features, and calculate DREAMSIM (**DSIM**; Fu et al., 2023), a learned perceptual similarity metric designed to align with human judgments.

**Results** Table 3 presents the system-level metric scores for conditional evaluation. The perceptual similarity metrics consistently demonstrate that MULTIMAT (Graph) achieves the highest fidelity to input images, with MULTIMAT (Mixed) performing second-best and VLMATERIAL (SBS) ranking last. For example, DREAMSIM scores are 36.609, 34.922, and 31.344, respectively, a ranking that mirrors our unconditional evaluation results. Parameter optimization yields substantial improvements in perceptual similarity, with MULTIMAT$^+$ (Graph) and MULTIMAT$^+$ (Mixed) showing average gains of 6% and 8%, respectively. In contrast, VLMATERIAL$^+$ (SBS) exhibits minimal improvement (only 1%), suggesting its outputs deviate too far from the input for parameter optimization to be effective. Interestingly, while parameter optimization improves perceptual similarity, KID scores increase. This could occur because optimization aligns outputs more closely with the test set, which represents only a subset of the training distribution, potentially increasing distance from the full distribution. Nevertheless, both MULTIMAT and MULTIMAT$^+$ variants outperform VLMATERIAL (SBS) and VLMATERIAL$^+$ (SBS) on KID by over 10pp, respectively. The remaining metrics reinforce trends from unconditional evaluation. ROUGE-L scores do not exceed 2% (indicating minimal memorization), and MULTIMAT (Mixed) produces the fewest errors. Overall, MULTIMAT (Graph) and its optimized variant, MULTIMAT$^+$ (Graph), deliver the strongest performance across metrics.

## 6.3 HUMAN EVALUATION

To corroborate our automatic evaluation results, we conduct a human evaluation. We employ comparative annotation (Thurstone, 1927) and focus on the image reconstruction/inverse rendering use case, which allows for intuitive human assessment (cf. Figure 6). Annotators receive triplets of rendered generated materials from VLMATERIAL$^+$ (SBS), MULTIMAT (Mixed), and MULTIMAT (Graph) and identify which output best and least resembles the input image. Following Hu et al.

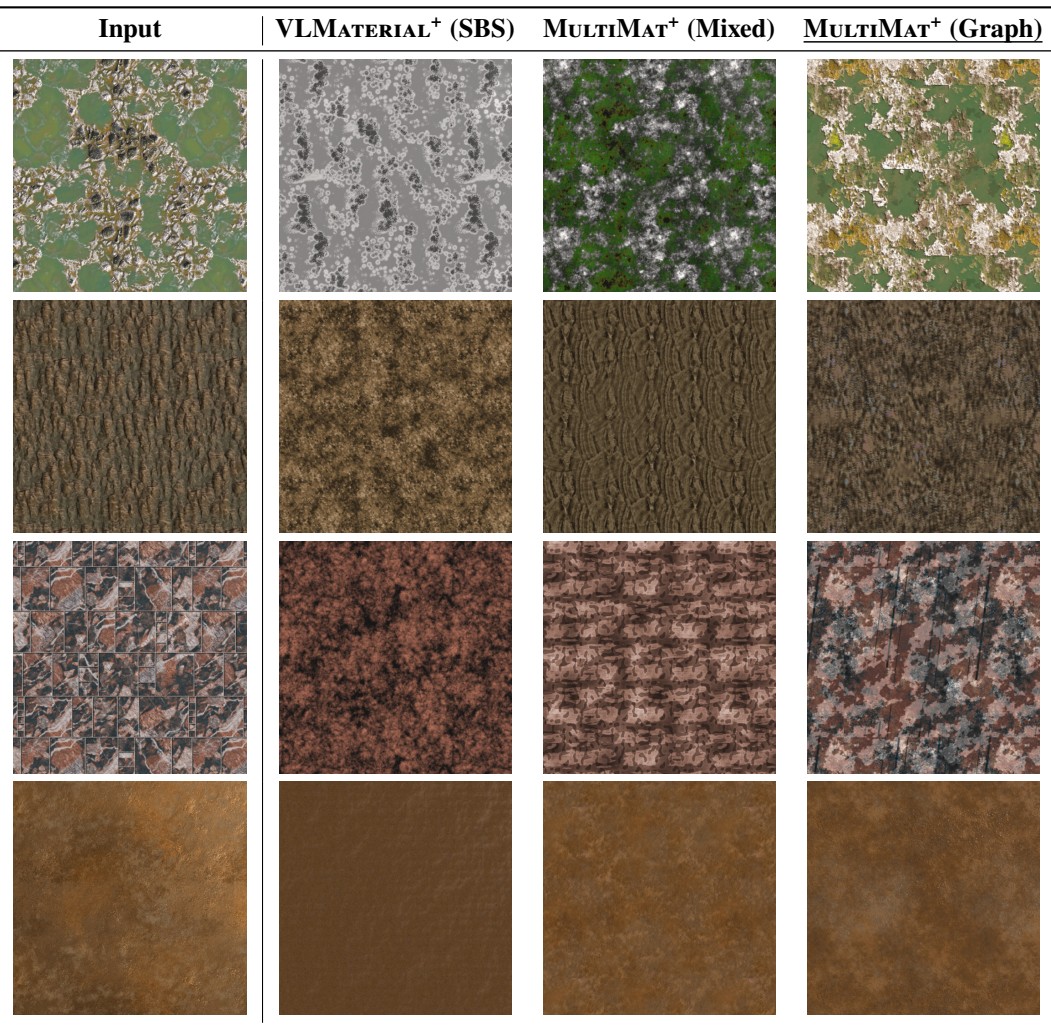

Figure 6: Qualitative examples for inverse procedural material modeling following the setup of our human evaluation in §6.3. The leftmost column shows input materials from graphs filtered during preprocessing, making these particularly challenging test cases. MultiMat⁺ (Mixed) consistently outperforms VLMaterial⁺ (SBS), while MultiMat⁺ (Graph) achieves the best results overall. Additional examples, including failure cases, are provided in Appendix A.

(2023); Li et al. (2025a), we generate multiple programs ($N = 40$) per model and image, selecting the result with the highest DreamSim score as the final candidate. We test 33 input materials from graphs filtered during preprocessing (e.g., due to excessive length), which represent particularly challenging cases. Eight expert annotators with extensive procedural material experience assess each triplet in randomized order. As shown in Figure 5, annotators rank VLMaterial⁺ (SBS) considerably lower than MultiMat⁺ (Mixed) and MultiMat⁺ (Graph), and prefer MultiMat⁺ (Graph) over MultiMat⁺ (Mixed). These findings align with our automatic evaluation rankings and demonstrate our approach's effectiveness in generating perceptually similar materials.

## 7    Analysis & Discussion

Our comparisons demonstrate that model performance improves steadily as the degree of graph visualization increases, with MultiMat (Graph) achieving the highest performance overall (cf. Tables 2 & 3; Figure 5). This finding aligns with how humans interact with procedural materials—through visual node graph interfaces—and validates established UX design principles in this domain.

The qualitative examples in Figure 6 further illustrate this trend, with VLMATERIAL⁺ (SBS) struggling to generate faithful outputs, indicating that purely text-based approaches are not ideal for expressive node graph systems like Designer. This limitation persists even with more powerful base models, as our unconditional generation experiments confirm. Beyond architectural improvements, our tree search algorithm enables more efficient graph generation; without it, models may have to resort to sampling complete outputs for validation (the inference approach used by previous methods), which is expen-

| Models | Deletion↓ | Conversion↓ |
|---|---|---|
| VLMATERIAL (SBS) | 2.71 | 12.26 |
| MULTIMAT (Mixed) | 1.18 | **3.51** |
| MULTIMAT (Graph) | **1.1** | 6.49 |

Table 4: Percentage of nodes repaired through parameter deletion or conversion node insertion in our unconditional and conditional evaluations. Bold and underlined values indicate the best and second-best scores for each metric column, respectively. Arrows indicate metric directionality. Our MULTIMAT models require the least amount of repair.

sive. For instance, disabling tree search causes NER of VLMATERIAL (SBS) to deteriorate further from 14.846 to 33.953, highlighting how our search strategy can improve inference without further training. The impact of automatic error repair is more nuanced, as shown in Table 4. Only approximately 1% of nodes generated by MULTIMAT contain hallucinated parameters, and fewer than 6.5% require conversion. In contrast, VLMATERIAL exhibits nearly double the scores for both repair mechanisms. This difference demonstrates that VLMATERIAL requires considerably more repair than our models and supports our claim that our models possess a better understanding of graph structures. Notably, since corrections are not fed back to the models, these results reflect their intrinsic generation capabilities.

## 8 CONCLUSION

We present MULTIMAT, a multimodal program synthesis framework and model suite that generates procedural materials by incorporating visual feedback throughout the generation process. Our key insight is that procedural material graphs are inherently visual-spatial programs, and treating them as such leads to substantial improvements over text-only approaches. By conditioning on visual intermediate states—either interleaved with text (mixed conditioning) or as complete graph visualizations (graph conditioning)—our models achieve consistent improvements over text-only baselines. Our incremental tree search algorithm further enhances generation efficiency by validating nodes as they are created and backtracking upon errors. While we demonstrate MULTIMAT specifically for procedural material synthesis, we hope its general principles will inspire further research at the intersection of computer graphics, program synthesis, and multimodal AI.

**Future Work** The development of procedural material graph synthesis approaches is currently constrained by limited training data availability. We plan to address this challenge through self-learning techniques (He et al., 2020; Wei et al., 2021) that leverage our unconditional models to generate synthetic supervised training data by rendering outputs and subsequently training conditional models on this expanded data. Additionally, we aim to develop a unified model trained across multiple node graph systems to investigate potential transfer learning benefits (Pan & Yang, 2010). Beyond methodological advances, our models offer promising practical applications: conditional models could extract material graphs directly from photographic regions, while unconditional models could power intelligent auto-completion features in user interfaces. Furthermore, our methodology naturally extends to related domains such as vector graphics synthesis (Wu et al., 2023; Polaczek et al., 2025; Rodriguez et al., 2025; Yang et al., 2025b), where visual editing interfaces are similarly prevalent.

**Limitations** Although our models and baselines use the same or similar base models, they generate graphs in fundamentally different ways, resulting in considerable differences in training efficiency. Text-only models like VLMATERIAL can process entire graphs as single training examples, whereas MULTIMAT must adapt the visual context for each individual node, effectively processing training examples one node at a time. This difference leads to much longer training times: while VLMATERIAL completes training in a few hours on $8 \times$ A100 80GB GPUs, MULTIMAT models require several days on the same hardware despite being trained on a comparable number of tokens. Nevertheless, since the amount of procedural materials is very small (regardless of the dataset), training times remain within acceptable bounds in absolute terms, despite the relative differences between methods. Additionally, both approaches achieve a more similar throughput during inference.

ETHICS STATEMENT

We ensure that all procedural materials collected for model training are properly licensed and explicitly permit such usage, thereby preventing any copyright infringement. In adherence to this principle, we specifically exclude Substance 3D Community Assets (Adobe, 2025b) from our training data due to licensing restrictions. While we acknowledge the use of generative models in preparing this manuscript, their application is strictly limited to writing assistance, such as paraphrasing, spell checking, and synonym suggestions.

ACKNOWLEDGEMENTS

We thank the Adobe Substance 3D team for providing access to the Substance 3D Assets Repository and the Substance Automation Python API. We also thank our annotators for their valuable time. This work was conducted while the first author was an intern at Adobe Research, France.

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

# A  ADDITIONAL EXAMPLES

In Figure 7 we provide additional qualitative examples. MULTIMAT$^+$ (Mixed) consistently surpasses VLMATERIAL$^+$ (SBS), while MULTIMAT$^+$ (Graph) demonstrates the strongest results overall. Figure 8 complements Figures 6 & 7 by showcasing failure cases where our models struggle to produce faithful outputs, though notably, the outputs from MULTIMAT$^+$ (Graph) and MULTIMAT$^+$ (Mixed) still demonstrate superior representation of the input compared to VLMATERIAL (SBS). Beyond these conditional generation examples, Figure 9 presents unconditional samples generated by MULTIMAT (Graph), which exhibit high visual quality with realistic material properties. Adjacent to these rendered materials, we visualize their underlying material graphs in the same format used as model input. In Figure 10, we show a graph in COMPACTSBS representation to give an impression of the structure of our format.

| Input | VLMATERIAL⁺ (SBS) | MULTIMAT⁺ (Mixed) | MULTIMAT⁺ (Graph) |
|---|---|---|---|

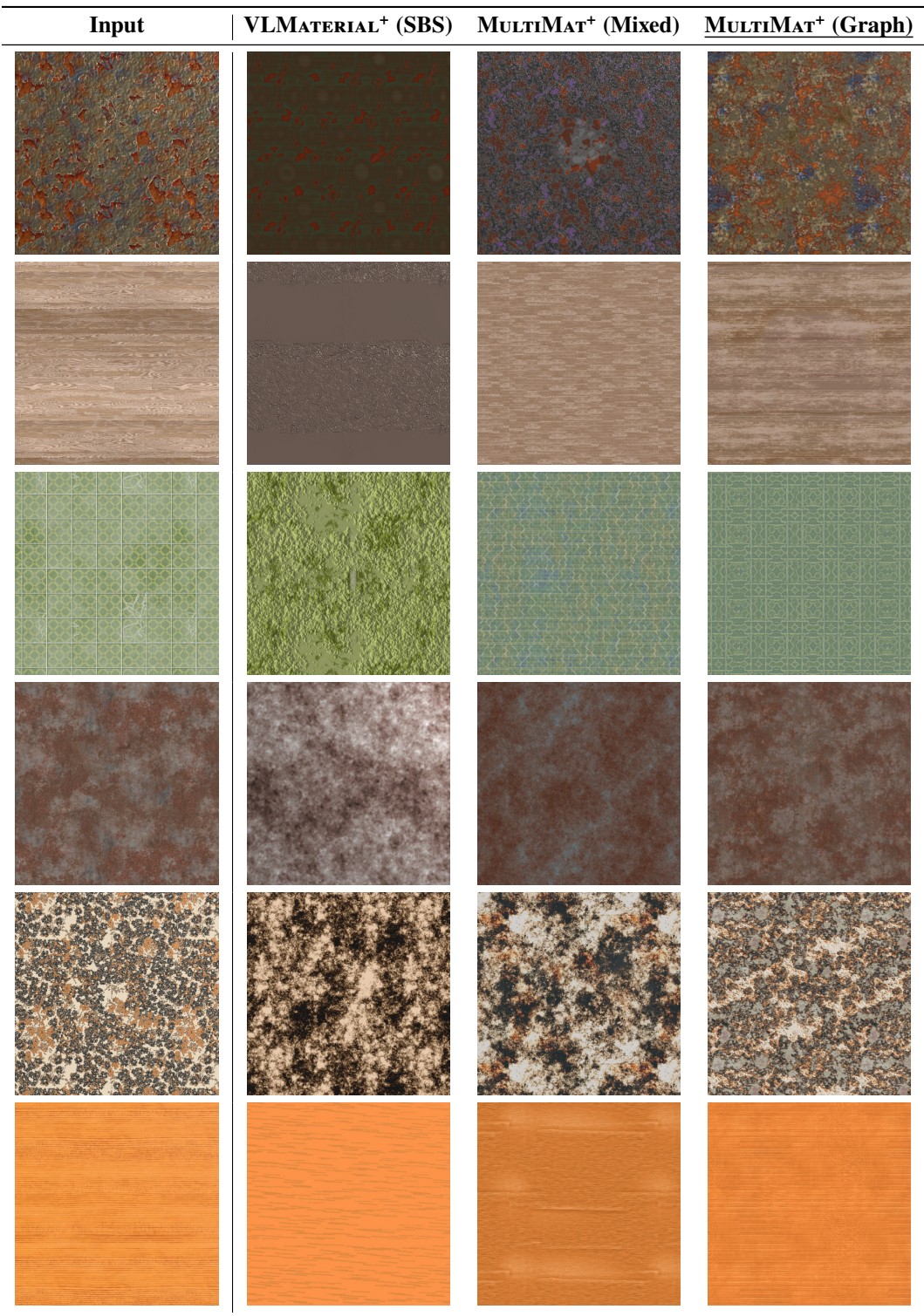

Figure 7: Additional qualitative examples for inverse procedural material modeling following the setup of our human evaluation in §6.3. The leftmost column shows input materials from graphs filtered during preprocessing, making these particularly challenging test cases. MULTIMAT⁺ (Mixed) consistently outperforms VLMATERIAL⁺ (SBS), while MULTIMAT⁺ (Graph) achieves the best results overall.

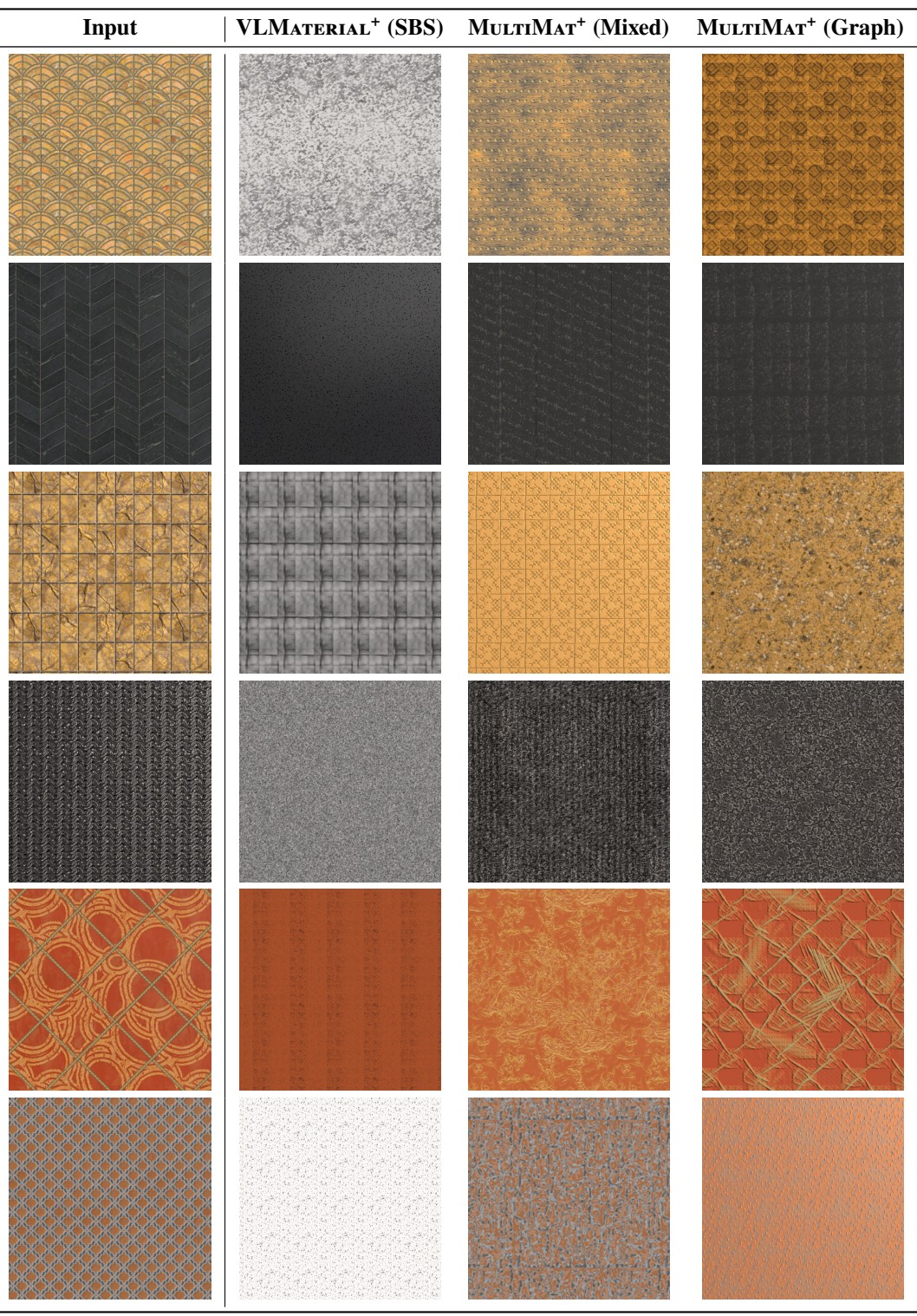

Figure 8: Representative failure cases from the same challenging subset in Figures 6 & 7. All models struggle to reproduce the intricate patterns in these examples, though MULTIMAT⁺ (Graph) and MULTIMAT⁺ (Mixed) still outperform VLMATERIAL⁺ (SBS).

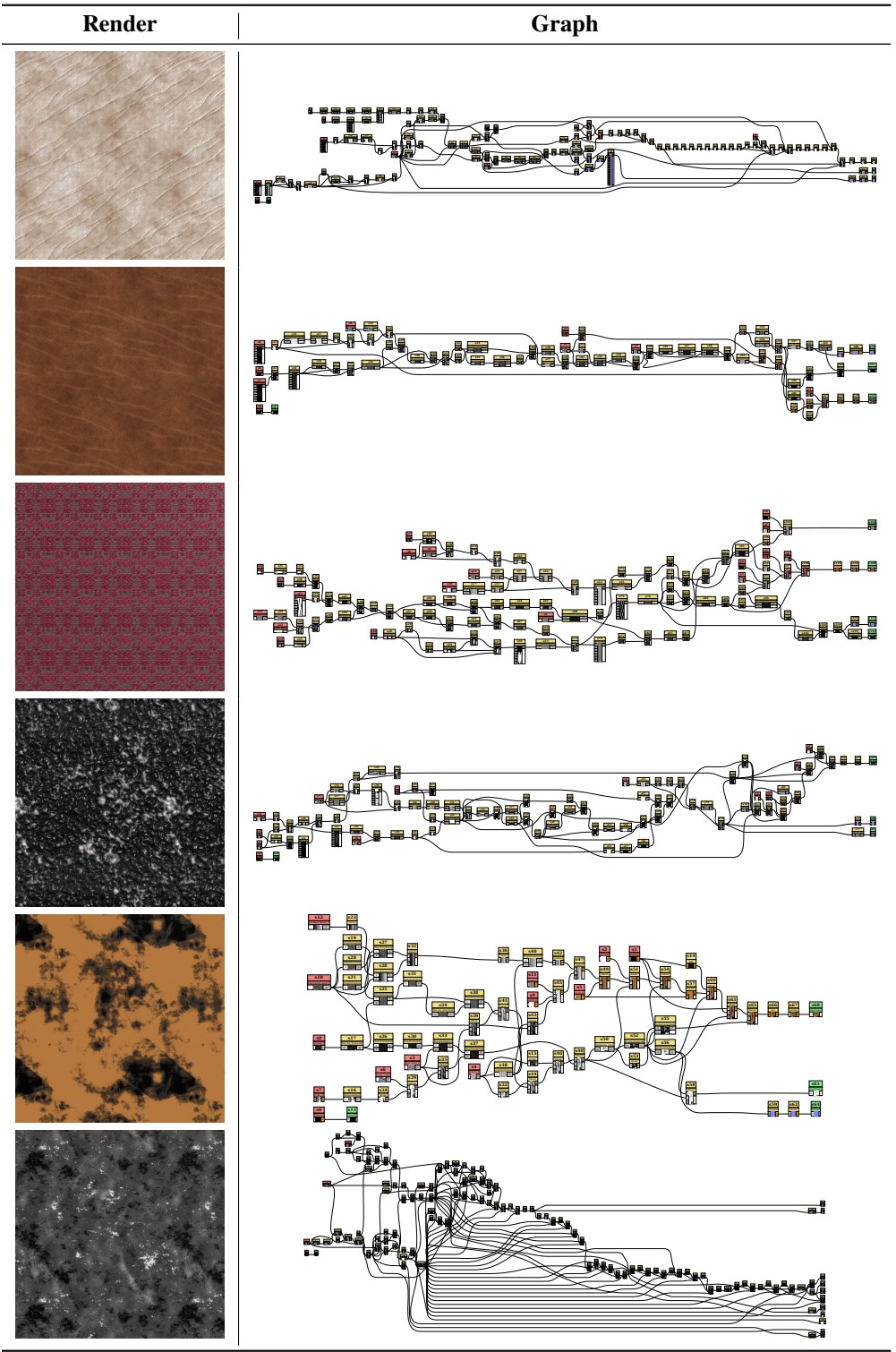

Figure 9: Example materials generated unconditionally by MULTIMAT (Graph), shown alongside their corresponding procedural graphs.

```
variables:
  contrast: 0.0
  fabric_color: [0.94, 0.79, 0.69]
  fabric_metallic: 0.0
  fabric_roughness: 0.23
  height_position: 0.5
  height_range: 1.0
  hue_shift: 0.0
  luminosity: 0.5
  normal_format: 0
  normal_intensity: 0.5
  saturation: 0.5
s0:
  function: uniform
  params:
    absolute:
      colorswitch: false
      outputcolor:
        f0:
          function: get_float1
          params:
            get_float1: fabric_metallic
s1:
  function: uniform
  params:
    absolute:
      colorswitch: false
      outputcolor:
        f0:
          function: get_float1
          params:
            get_float1: fabric_roughness
      outputsize: [4, 4]
s2:
  function: uniform
  params:
    absolute:
      colorswitch: false
      outputcolor: [0.5, 0.5, 0.5, 1.0]
      outputsize: [4, 4]
s3:
  function: uniform
  params:
    absolute:
      outputcolor:
        f0:
          function: get_float3
          params:
            get_float3: fabric_color
        f1:
          function: const_float1
          params:
            const_float1: 1.0
        f2:
          function: vector4
          connections:
            componentsin: f0
            componentslast: f1
      outputsize: [4, 4]
s4:
  function: tile_generator
  dependency:
    sbs://pattern_tile_generator.sbs
  params:
    absolute:
      pattern: 4
      scale: 2.0
      interstice: [0.64, 0.0, 0.0, 0.0]
      blending_mode: 2
      rotation: 0.05
      luminance_random: 0.55
      y_amount: 200
      x_amount: 150
      position_offset: 0.5
      vertical_offset: true
s5:
  function: fractal_sum_base_2
  dependency:
    sbs://noise_fractal_sum_base.sbs
s6:
  outputs:
    metallic: RGBA
  connections:
    inputNodeOutput:
      node: s0
      id: output
s7:
  function:
    multi_directional_warp_grayscale
  dependency:
    sbs://multi_directional_warp.sbs
  connections:
    input:
      node: s4
      id: output
    intensity_input:
      node: s5
      id: output
  params:
    absolute:
      intensity: 3.25
s8:
  function: transformation
  connections:
    input1:
      node: s7
      id: output
  params:
    absolute:
      offset: [0.38, 0.54]
      matrix22: [-1.0, 0.0, 0.0, 1.0]
s9:
  function: blend
  connections:
    destination:
      node: s7
      id: output
    source:
      node: s8
      id: output
  params:
    absolute:
      blendingmode: MAX
s10:
  function: safe_transform_grayscale
  dependency: sbs://safe_transform.sbs
  connections:
    input:
      node: s9
      id: output
  params:
```

```
      absolute:
        rotation: 0.25
        tile: uU_vV
s11:
  function: blend
  connections:
    destination:
      node: s2
      id: output
    source:
      node: s10
      id: output
  params:
    absolute:
      blendingmode: ADD
      opacitymult: 0.2
      format: 1
    parent:
      outputsize: [0, 0]
s12:
  function: levels
  connections:
    input1:
      node: s10
      id: output
  params:
    absolute:
      levelinlow: [0.02, 0.02, 0.02, 0.0]
      levelinhigh: [0.95, 0.95, 0.95, 1.0]
      leveloutlow: [1.0, 1.0, 1.0, 1.0]
      levelouthigh: [0.0, 0.0, 0.0, 0.0]
      levelinmid: [0.41, 0.41, 0.41, 0.5]
s13:
  function: highpass_grayscale
  dependency: sbs://highpass.sbs
  connections:
    Source:
      node: s10
      id: output
  params:
    absolute:
      Radius: 0.1
s14:
  function: normal
  connections:
    input1:
      node: s11
      id: output
  params:
    absolute:
      intensity:
        f0:
          function: get_float1
          params:
            get_float1: normal_intensity
        f1:
          function: const_float1
          params:
            const_float1: 3.0
        f2:
          function: mul
          connections:
            a: f0
            b: f1
      inversedy:
        f0:
          function: get_integer1
          params:
            get_integer1: normal_format
        f1:
          function: const_int1
          params:
            const_int1: 1
        f2:
          function: eq
          connections:
            a: f0
            b: f1
      input2alpha: false
s15:
  function: histogram_range
  dependency: sbs://histogram_range.sbs
  connections:
    input:
      node: s11
      id: output
  params:
    absolute:
      range:
        f0:
          function: get_float1
          params:
            get_float1: height_range
      position:
        f0:
          function: get_float1
          params:
            get_float1: height_position
s16:
  function: blend
  connections:
    destination:
      node: s1
      id: output
    source:
      node: s12
      id: output
  params:
    absolute:
      blendingmode: SCREEN
      opacitymult: 0.15
    parent:
      outputsize: [0, 0]
s17:
  function: levels
  connections:
    input1:
      node: s13
      id: Highpass
  params:
    absolute:
      levelinlow: [0.33, 0.33, 0.33, 0.0]
      levelinhigh: [0.61, 0.61, 0.61, 1.0]
      leveloutlow: [1.0, 1.0, 1.0, 1.0]
      levelouthigh: [0.0, 0.0, 0.0, 0.0]
s18:
  outputs:
```

```
    normal: RGBA
    inputNodeOutput:
      node: s14
      id: output
s19:
  outputs:
    height: RGBA
    inputNodeOutput:
      node: s15
      id: output
s20:
  outputs:
    roughness: RGBA
    inputNodeOutput:
      node: s16
      id: output
s21:
  function: blend
  connections:
    destination:
      node: s3
      id: output
    opacity:
      node: s17
      id: output
    source:
      node: s3
      id: output
  params:
    absolute:
      blendingmode: MULTIPLY
      opacitymult: 0.35
    parent:
      outputsize: [0, 0]
s22:
  function: hsl
  connections:
    input1:
      node: s21
      id: output
  params:
    parent:
      hue:
        f0:
          function: get_float1
          params:
            get_float1: hue_shift
        f1:
          function: const_float1
          params:
            const_float1: 0.5
        f2:
          function: mul
          connections:
            a: f0
            b: f1
      saturation:
        f0:
          function: get_float1
          params:
            get_float1: saturation
      luminosity:
        f0:
          function: get_float1
          params:
            get_float1: luminosity
s23:
  function: levels
  connections:
    input1:
      node: s22
      id: output
  params:
    parent:
      levelinlow:
        f0:
          function: const_float1
        f1:
          function: const_float1
          params:
            const_float1: 0.5
        f2:
          function: get_float1
          params:
            get_float1: contrast
        f3:
          function: max
          connections:
            a: f2
            b: f0
        f4:
          function: mul
          connections:
            a: f3
            b: f1
        f5:
          function: vector2
          connections:
            componentsin: f4
            componentslast: f4
        f6:
          function: vector2
          connections:
            componentsin: f4
            componentslast: f0
        f7:
          function: vector4
          connections:
            componentsin: f5
            componentslast: f6
      levelinhigh:
        f0:
          function: const_float1
          params:
            const_float1: 1.0
        f1:
          function: const_float1
          params:
            const_float1: 0.5
        f2:
          function: const_float1
        f3:
          function: get_float1
          params:
```

```
      get_float1: contrast
        f4:
          function: max
          connections:
            a: f3
            b: f2
        f5:
          function: mul
          connections:
            a: f4
            b: f1
        f6:
          function: sub
          connections:
            a: f0
            b: f5
        f7:
          function: vector2
          connections:
            componentsin: f6
            componentslast: f6
        f8:
          function: vector2
          connections:
            componentsin: f6
            componentslast: f0
        f9:
          function: vector4
          connections:
            componentsin: f7
            componentslast: f8
      leveloutlow:
        f0:
          function: const_float1
        f1:
          function: const_float1
          params:
            const_float1: 0.5
        f2:
          function: get_float1
          params:
            get_float1: contrast
        f3:
          function: min
          connections:
            a: f2
            b: f0
        f4:
          function: abs
          connections:
            a: f3
        f5:
          function: mul
          connections:
            a: f4
            b: f1
        f6:
          function: vector2
          connections:
            componentsin: f5
            componentslast: f5
        f7:
          function: vector2
          connections:
            componentsin: f5
            componentslast: f0
        f8:
          function: vector4
          connections:
            componentsin: f6
            componentslast: f7
      levelouthigh:
        f0:
          function: const_float1
          params:
            const_float1: 1.0
        f1:
          function: const_float1
          params:
            const_float1: 0.5
        f2:
          function: const_float1
        f3:
          function: get_float1
          params:
            get_float1: contrast
        f4:
          function: min
          connections:
            a: f3
            b: f2
        f5:
          function: abs
          connections:
            a: f4
        f6:
          function: mul
          connections:
            a: f5
            b: f1
        f7:
          function: sub
          connections:
            a: f0
            b: f6
        f8:
          function: vector2
          connections:
            componentsin: f7
            componentslast: f7
        f9:
          function: vector2
          connections:
            componentsin: f7
            componentslast: f0
        f10:
          function: vector4
          connections:
            componentsin: f8
            componentslast: f9
s24:
  outputs:
    baseColor: RGBA
  connections:
    inputNodeOutput:
      node: s23
      id: output
```

Figure 10: Complete example of a graph in CompactSBS format. This listing shows the full representation of the material partially illustrated in Figure 3.

