# OpenReview forum: "MultiMat: Multimodal Program Synthesis for Procedural Materials using Large Multimodal Models"
_ICLR.cc/2026/Conference — ICLR 2026 Poster_

### Official Review · Reviewer_Zyod · 2025-10-22

**Soundness:** 2
**Presentation:** 2
**Contribution:** 2
**Rating:** 2
**Confidence:** 2

**Summary:**

This paper introduces MultiMat, a multimodal program synthesis framework that generates procedural material node graphs using large multimodal models. Unlike previous text-only approaches, MultiMat incorporates visual feedback by processing both textual representations and visualizations of intermediate graph states. The authors propose two conditioning methods (mixed and graph-based) and implement a constrained tree search algorithm for efficient inference. Experiments on a dataset of 6,878 production-quality Adobe Substance Designer materials demonstrate superior performance over text-only baselines in both unconditional and conditional (inverse rendering) generation tasks.

**Strengths:**

1: The key insight of treating procedural material graphs as visual-spatial programs is intuitive and well-motivated, aligning with how human artists interact with these tools.

2: The collection of 6,878 materials with complete feature set support represents a significant improvement over previous datasets.

**Weaknesses:**

1: The authors acknowledge that MultiMat requires "several days" on 8×A100 GPUs compared to "a few hours" for text-only baselines. This ~10-20× difference in training time is a significant practical limitation that deserves more discussion about potential optimizations.

2:  All training and testing use the same material engine (Substance Designer). How would the approach transfer to other procedural material systems like Blender's shader nodes?

3: Lack of ablation study. What is the contribution of each component (visual feedback, tree search, error repair)? Ablation experiments would strengthen the paper.

**Questions:**

The paper relies entirely on automatic metrics. Given that visual quality is subjective, human evaluation comparing outputs from different methods would be valuable.

---

> ### Author Response · Authors · 2025-11-18
> **Response to Reviewer Zyod**
>
> We thank the reviewer for their time and are happy to address any concerns. Please also check our general response (W1, W2 & Q1).
>
> > W3: Lack of ablation study. What is the contribution of each component (visual feedback, tree search, error repair)? Ablation experiments would strengthen the paper.
>
> We appreciate the reviewer’s suggestion and agree that these ablations are valuable. We would like to clarify that the paper already addresses these points. Removing visual feedback effectively reduces our approach to VLMaterial, which our method surpasses. Additionally, Section 7 analyzes the impact of disabling tree search, where we observe a substantial increase in the Node Error Ratio. We will highlight this findings more clearly in the revision.
>
> We hope we have addressed the reviewer's concerns in this response and the general response, and we kindly ask them to reconsider their assessment.

---

> > ### Comment · Reviewer_Zyod · 2025-11-20
> >
> > Many thanks for the authors' reply.
> >
> > Regarding W3, Section 7 is too short and does not constitute a proper ablation study. The only quantitative result is that disabling tree search worsens NER on VLMaterial (SBS) from 14.846 to 33.953. A more systematic study is required; since the proposed pipeline involves both an LLM and procedural generation, assessing the specific contribution of each design component more quantitatively is essential.
> >
> > User study: Results are nice. Thank you for conoducting it, this will be helpful to understand the performance improvements over previous models.
> >
> > Regarding W1 and W2, my concerns about efficiency and the single procedural system are shared by other reviewers as well. Although the general response touches on these points, it lacks actual experiments showing how to resolve them. Therefore, I remain unconvinced and I'm afraid I cannot recommend accepting this paper at this stage.

---

> > > ### Author Response · Authors · 2025-11-25
> > > **Update to Reviewer Zyod**
> > >
> > > We thank the reviewer for taking the time to respond to our rebuttal. We would like to emphasize that the requested quantitative assessment of each design component's specific contributions is already included in our paper. As the reviewer correctly notes, this information is not in Section 7; rather, it can be found as part of our main experiments in Section 6. By comparing VLMaterial with MultiMat (Mixed) and MultiMat (Graph), we evaluate the gradual shift from purely auto-regressive graph generation toward iterative node generation through both quantitative automatic and human evaluation. This comparison also assesses how increasing degrees of graph visualization impact performance. These analyses clearly demonstrate the sources of MultiMat's performance improvements, which is why we believe our paper already addresses this question sufficiently. We have clarified this point in our revision.
> > >
> > > >Regarding W1 and W2, my concerns about efficiency and the single procedural system are shared by other reviewers as well
> > >
> > > While other reviewers also mention these points, they generally rate our work favorably, as these concerns are not critical (as explained in our general response). Although targeting a single procedural material system is standard practice, our approach generates an intermediate format that is then transpiled to Substance Designer. This architecture makes it straightforward to implement additional backends for other systems (e.g., Blender), since all material graph systems provide the necessary linear algebra operators to replicate any node across platforms. As stated in our general response, we consider this an engineering rather than research challenge. However, if the reviewer wishes, we would be happy to include an example in the supplementary materials.

---

> > > > ### Comment · Reviewer_Zyod · 2025-11-25
> > > >
> > > > Thanks to the authors for the additional response and clear explanations. While some of my concerns remain, I have updated my score based on the changes in the response.

---

### Official Review · Reviewer_tCBc · 2025-10-30

**Soundness:** 3
**Presentation:** 3
**Contribution:** 3
**Rating:** 6
**Confidence:** 2

**Summary:**

The paper proposes MultiMat, a multimodal framework for synthesizing procedural material node graphs. Unlike prior text-only methods, MultiMat leverages both textual and visual representations during generation, integrating visual feedback to better mimic human material design workflows. It introduces (1) a multimodal feedback loop using intermediate graph visualizations, (2) an incremental tree search with automatic error repair for valid program generation, and (3) a new dataset of production-quality materials from Adobe Substance Designer.

**Strengths:**

1. Novel multimodal approach combining visual and textual cues for program synthesis.
2. Strong empirical results, showing clear gains in KID, CLIP, and DreamSim metrics.
3. Creation of a large, comprehensive dataset supporting full Substance Designer features.

**Weaknesses:**

1. Training inefficiency: requires per-node multimodal context updates, leading to much longer training times.
2. Evaluation primarily limited to one domain (Substance Designer), with generalization to other systems unclear.

**Questions:**

1. How well does MultiMat generalize to unseen node types or non-Designer graph systems?
2. Could the incremental tree search approach be applied to other program synthesis domains (e.g., 3D modeling or animation)?

---

> ### Author Response · Authors · 2025-11-18
> **Response to Reviewer tCBc**
>
> We thank the reviewer and are glad that they generally like our work. We will now answer raised questions and weaknesses. Please also check our general response (W1, W2).
>
> > Q1: How well does MultiMat generalize to unseen node types or non-Designer graph systems?
>
> As a data-driven system, MultiMat cannot generate unseen node types. However, if conditioned on an unseen node type (for example, when working with a partially generated graph containing such nodes), MultiMat would likely understand the node's function through its visualization. Consequently, MultiMat has a much higher likelihood of correctly continuing graphs containing unseen nodes compared to our text-based baselines.
>
> > Q2: Could the incremental tree search approach be applied to other program synthesis domains (e.g., 3D modeling or animation)?
>
> MultiMat's approach could be straightforwardly applied to other domains where programs are compositional (i.e., where intermediate states can be executed). This includes 3D modeling or animation but also extends to vector graphics. We see many opportunities that could be explored in future work.

---

> > ### Comment · Reviewer_tCBc · 2025-11-19
> >
> > Thank you for the authors' response. I'll maintain my positive score.

---

### Official Review · Reviewer_d96n · 2025-11-01

**Soundness:** 2
**Presentation:** 3
**Contribution:** 2
**Rating:** 4
**Confidence:** 3

**Summary:**

This paper presents MultiMat, a framework for synthesizing procedural material graphs using large multimodal models on both visual and textual data. Arguing that material creation is an inherently visual-spatial task, the authors show that incorporating visual intermediate states improves results over text-only methods. The system is validated on a new dataset from Adobe Substance Designer through extensive experiments, including unconditional and image-guided synthesis, ablations, and comparisons to recent text-only and multimodal baselines.

**Strengths:**

1. The paper's central contribution is a multimodal synthesis paradigm that incorporates visual feedback of intermediate states during graph generation. This method addresses a key limitation in text-only approaches by more closely aligning with human creative workflows.

2. A new, comprehensive dataset of production-level procedural materials is collected and used for evaluation. The experiments are rigorous, benchmarking against strong baselines on both unconditional and conditional synthesis tasks and demonstrating superior performance over text-only models across multiple metrics.

3. The work includes systematic ablation studies and an analysis of failure cases, which add credibility by clarifying the model's capabilities and current limitations. The paper also presents practical error recovery strategies, such as incremental tree search and automatic repair.

4. The exposition is clear and well-supported by figures that effectively visualize the model architecture, workflows, and experimental results. The inclusion of concrete output examples, such as visual graphs and code listings, helps to illustrate the system's real-world performance.

**Weaknesses:**

- The paper's contextualization is limited by its failure to engage with recent, directly relevant literature. The Related Work section omits several key advances in multimodal program synthesis (e.g., [1], [2], [3] ) and recent visual programming benchmarks.

- The methodology lacks mathematical formalism, which hinders reproducibility and theoretical interpretation. Crucial elements like the multimodal training objective, loss functions, negative sampling strategy, and the method for combining image and text embeddings are not precisely specified. Algorithmic descriptions, such as the backtracking process, also lack the clarity of formal pseudocode.

- Missing citations to several directly relevant works: As detailed below, the references omit numerous recent multimodal and vision-language approaches to program synthesis and visual programming benchmarks [4][5].

- The experimental analysis is not sufficiently thorough. The ablation studies are limited and do not provide quantitative evidence for the impact of individual components like automatic error repair. The paper also needs a clearer discussion on the fairness of baseline comparisons, a deeper analysis of the trade-off between efficiency and accuracy, and more comprehensive qualitative comparisons.
Please provide some rendered videos.

- Claims of the method's generality are conjectural, as the entire pipeline and dataset are specific to Substance Designer. The paper provides no experimental evidence or case studies to demonstrate the framework's adaptability to other graph-based creative environments.

[1]. Optimal Neural Program Synthesis from Multimodal Specifications

[2].  Synthesis of Programs from Multimodal Datasets

[3]. Program Synthesis Benchmark for Visual Programming

[4]. JarvisArt: Liberating Human Artistic Creativity via an Intelligent Photo Retouching Agent

[5]. Visual Planning: Let's Think Only with Images

**Questions:**

- Could you offer a more formal definition of the multimodal objective function, with equations showing how image and text data are fused during generation? It would be helpful to know if attention is shared and whether the loss is contrastive or generative.

- Since the evaluation is confined to Substance Designer, could a case study on a different visual programming language be provided to substantiate the claims of generality, even if the experiment is on a smaller scale?

- For the retrained baselines like VLMaterial and MatFormer, please clarify which features were omitted or unsupported. How might these omissions impact the fairness of the performance comparisons?

---

> ### Author Response · Authors · 2025-11-18
> **Response to Reviewer d96n**
>
> We thank the reviewer for their thorough review and are happy to address weaknesses and questions. Please also check our general response (W2, parts of W4, W5, Q2).
>
> > W1: The paper's contextualization is limited by its failure to engage with recent, directly relevant literature. The Related Work section omits several key advances in multimodal program synthesis (e.g., [1], [2], [3] ) and recent visual programming benchmarks.
>
> We agree that the papers [1,2] represent important advances in program synthesis, and we have happily included them in our related work section, but they are only tangentially related to our work. To clarify, in our paper, *multimodal* in multimodal program synthesis refers to two distinct aspects: (i) *visual* inputs and outputs to and from textual programs, and (ii) programs that *themselves* combine visual and textual elements (i.e., our graph and mixed representations) which is a novel contribution. The cited papers [1,2] use the term multimodal differently, i.e., in their context, multimodal synthesis refers to systems where users provide both a logical specification and a natural language description. We acknowledge that the term multimodal is overloaded, which may have caused this confusion. We have clarified this distinction in the updated version of our paper. Paper [3] relates only distantly to aspect (ii), as it presents inputs as images. Nevertheless, we have gladly added it to our related work section.
>
> > W3: Missing citations to several directly relevant works: As detailed below, the references omit numerous recent multimodal and vision-language approaches to program synthesis and visual programming benchmarks [4][5].
>
> This point raises a similar concern about missing citations for visual program synthesis as in W1. We note that the cited papers [4,5] do not address program synthesis; however, we agree that they are relevant to our work and have included them in our updated paper for completeness. If the reviewer believes we have overlooked any other key contributions, particularly in our literature review in Section 2, we would be happy to incorporate them as well.
>
> > W4: The experimental analysis is not sufficiently thorough. The ablation studies are limited and do not provide quantitative evidence for the impact of individual components like automatic error repair. The paper also needs a clearer discussion on the fairness of baseline comparisons [...].
> > Q3: For the retrained baselines like VLMaterial and MatFormer, please clarify which features were omitted or unsupported. How might these omissions impact the fairness of the performance comparisons?
>
> We agree that a clear discussion on baseline comparison fairness is important. Our retraining of baselines is necessary to ensure fair comparisons with our models, so that they use the same base models and node graph system/training data. Our experiments clearly demonstrate our approach's strengths. Notably, we did not omit any features from the base models during retraining; quite contrary, we build them on more recent and stronger base models than those in the original papers. We also equipped them with the same individual components, like tree search and error repair, to ensure fair comparison. We ablate such components in Section 7, where we show that disabling tree search substantially increases the Node Error Rate (NER), as expected. Following the reviewer's observation, we have clarified this in our updated version.
>
> Most of the weaknesses and questions raised were requests for clarification. We hope that these points address most of these concerns and kindly ask the reviewer to update their assessment.

---

> > ### Author Response · Authors · 2025-11-25
> > **Update to Reviewer d96n**
> >
> > We would like to notify the reviewer that we have updated our submission to include the requested rendered videos. As noted in our general response update, these videos demonstrate realistic light interactions and detailed surface properties of materials generated with our approach. Together with the expanded qualitative examples and our human evaluation study, we believe we have addressed the request for additional qualitative comparisons. We again kindly ask the reviewer to reconsider their assessment in light of these additions.

---

> ### Comment · Reviewer_d96n · 2025-11-26
>
> I appreciate the detailed response. I have read the other reviews and the authors' rebuttals.
>
> On the ablation study, while I see your point about `Section 6`, I still hold the view that comparing model variants (like Mixed vs. Graph) is effectively a system-level comparison rather than a true component ablation. As mentioned before, we need to see the isolated contribution of sub-modules. For example, measuring the exact gain from error repair independent of the generation strategy, to provide the quantitative clarity that is currently missing.
>
> Regarding the generality and the single-system issue (Substance Designer), I understand you view porting to other systems (like Blender) as merely an engineering task. However, since the paper claims broad applicability, we need empirical evidence to back that up, not just theoretical feasibility. I appreciate the offer to add a Blender example now, but I have to evaluate the work based on the results actually presented in the paper and the rebuttal so far.
>
> Since other reviewers (e.g., Reviewer Zyod) also flagged the efficiency and single-environment limitations, and the current experiments haven't fully cleared up these doubts, I’m still not convinced the results justify the broad claims. Also, the lack of sufficient qualitative demos leaves me with lingering concerns about the method's practical utility.
>
> Once again, I thank the authors for their hard work and comprehensive response and I remain open to further discussion.

---

> > ### Author Response · Authors · 2025-11-26
> > **Update to Reviewer d96n**
> >
> > We thank the reviewer for their continued constructive feedback, which has helped strengthen our work.
> >
> > > For example, measuring the exact gain from error repair independent of the generation strategy, to provide the quantitative clarity that is currently missing.
> >
> > Following this suggestion, we have expanded our analysis section with a deeper investigation of our error repair mechanism. Our findings reveal that only a small fraction of nodes generated by the MultiMat models require correction: approximately 1% of nodes contain hallucinated parameters, and fewer than 6.5% require type conversion. In contrast, VLMaterial exhibits nearly double these error rates for both repair mechanisms. This difference demonstrates that VLMaterial requires considerably more post-processing than our models, supporting our claim that our approach achieves superior understanding of procedural graph structures. Importantly, since corrections are not fed back to the models during generation, these results reflect their intrinsic generation capabilities. We believe these findings provide the quantitative clarity requested and strengthen our paper's contributions.
> >
> > > I understand you view porting to other systems (like Blender) as merely an engineering task. However, since the paper claims broad applicability, we need empirical evidence to back that up, not just theoretical feasibility.
> >
> > We respectfully note that our paper makes no claims beyond what our empirical findings support. In Section 3 (line 159), we explicitly state that our method targets Substance Designer. While we discuss potential future applications, these are clearly presented as directions for future work (as is standard practice in the field). If the reviewer believes we have inadvertently suggested broader applicability elsewhere in the text, we would appreciate specific references and would be happy to clarify any ambiguous statements.
> >
> > > Also, the lack of sufficient qualitative demos leaves me with lingering concerns about the method's practical utility.
> >
> > We would like to remind the reviewer that we have already added more qualitative examples to both the appendix and supplementary materials, including demonstration videos. While we acknowledge that our method has not yet reached human expert levels, it represents the current state-of-the-art in procedural material generation. Previous works such as VLMaterial and Matformer faced similar limitations yet were influental in their field. We believe our work represents a comparable advance in procedural material synthesis.

---

### Official Review · Reviewer_nJcx · 2025-11-03

**Soundness:** 3
**Presentation:** 3
**Contribution:** 3
**Rating:** 6
**Confidence:** 3

**Summary:**

This paper introduces MultiMat, which is a multimodal VLM trained for the accurate, efficient synthesis of 2D procedural material graphs (PMGs). MultiMat is the first procedural material approach that explicitly leverages both the textual and visual graph representations of PMGs to improve generation quality. The textual representation used in MultiMat is a new, simplified YAML-based description for PMGs, which is more readable and considerably (~80%) shorter than the native XML descriptions used by the underlying generation software (here, Adobe Substance Designer). During generation, MultiMat enforces an iterative, topologically-sorted generation procedure that verifies the graph after each node output, to ensure validity and allow efficient backtracking through constrained tree search. MultiMat also contributes a new PMG dataset based on the Substance Designer Assets Repository; this dataset is larger in size and scope than its predecessors.

**Strengths:**

This paper achieves SOTA performance across a variety of metrics on a challenging and impactful problem in computer graphics. Moreover, the paper achieves this by capitalizing on a remarkably simple observation about a piece of affiliate data that has traditionally been overlooked in PMG generation: the visual graph representation is crucial for human interaction with PMGs, so why shouldn't it be beneficial for VLMs as well?
Other domains beyond PMGs are likely to benefit from the notions in this work, such as reframing their data or adopting the iterative, verifiable graph generation with backtracking and error correction.
The paper is generally well written, including a substantive description of the domain, the related work and several comparisons to relevant baselines.

**Weaknesses:**

1. The paper does not specify details about the data format or the algorithms used to train the QWen model for MultiMat realization.
2. The baseline setup in 6.1 is unclear -- is VLMaterial embedded within the current paper's pipeline, as an exchange for the MultiMat block? I assumed that it was evaluated as a standalone VLMaterial model with full-graph generation, albeit trained over a different dataset. I was surprised, then, to read l. 412 describing the degradation of the NER score for VLMaterial(SBS) when tree search was turned off, since I never expected it to be on. A clearer description of the baseline setup would be very helpful.
3. Did you perform any ablation studies to test the efficacy/impact of your components? For example, does the fragmented node generation procedure affect the model's global reasoning ability over PMGs? This might be observable under the conditional generation task: is MultiMat is able to reproduce the desired target image equally well using iterative node generation vs single-shot graph generation?

**Questions:**

1. Why was it necessary/desirable to flatten hierarchical structures in the professional PMGs, and what (if any) are the ramifications of having done so? Do the substructures conflict with the tree search and/or the YAML representation? I ask because hierarchical subgroups are very useful for human designers, as they merge complex blocks into higher-level concepts whose internal details can be abstracted away once implemented; by the logic posited in this paper, I imagine that this would be similarly useful for the model once the graphs reach sufficient complexity (which is certainly possible within 128 nodes).
2. Rather than 1 node at a time, would it be possible for each iteration to generate a set of nodes all residing at the same topological sort depth? What would the potential advantages or challenges of such an approach be, relative to a single node?
3. Were the error patterns of Section 4.3 identified once (or a few times) through manual intervention, then hardcoded into the pipeline? Would there be any room to integrate feedback, such as online identification of failure modes and/or potential workarounds for them?
4. To my understanding, the iterative generation scheme provides information about the (partial) graph's validity and state visualization/description; however, it does not evaluated or provide feedback on e.g., whether the generated image is likely to lead to the target image under conditional generation. It seems like that would be an interesting extension though; would it be possible to evaluate and/or include any such information to shape the trajectory of the generation procedure? Do there exist any heuristics to judge the suitability of a partial solution for this domain?

-- Minor Comments --
Fig 1. The graph shown here emphasizes the possible complexity of professional PMGs, but does not illustrate the "intuitive" nature of these visualizations as you claim in e.g. l.68; it may be worth selecting a different graph here to make your point.

**Details Of Ethics Concerns:**

None.

---

> ### Author Response · Authors · 2025-11-18
> **Response to Reviewer nJcx**
>
> We thank the reviewer for their detailed review and are happy that they generally like our work. We are happy to address any weaknesses and questions. Please also check our general response (W1).
>
> > W2: The baseline setup in 6.1 is unclear -- is VLMaterial embedded within the current paper's pipeline, as an exchange for the MultiMat block? I assumed that it was evaluated as a standalone VLMaterial model with full-graph generation, albeit trained over a different dataset. [...]
>
> The reviewer's understanding is correct: we train and evaluate VLMaterial as a model with full-graph generation. However, during inference, we can still validate nodes as they are generated and roll back when errors are detected. This enables us to apply tree search (and error repair) to this model as well. We have clarified this point in our revision.
>
> > W3: Did you perform any ablation studies to test the efficacy/impact of your components? For example, does the fragmented node generation procedure affect the model's global reasoning ability over PMGs? This might be observable under the conditional generation task: is MultiMat is able to reproduce the desired target image equally well using iterative node generation vs single-shot graph generation?
>
> We believe our baseline comparisons address this question. When we replace iterative node generation with single-shot graph generation, our approach effectively becomes equivalent to VLMaterial. The progression from VLMaterial to MultiMat (mixed) to MultiMat (graph) represents a gradual shift from purely auto-regressive graph generation toward iterative node generation, resulting in progressively improved performance. We have clarified this point in our revision.
>
> > Q1: Why was it necessary/desirable to flatten hierarchical structures in the professional PMGs, and what (if any) are the ramifications of having done so? [...]
>
> We agree that subgraphs are very useful for humans, and models could likely benefit from such abstractions as well. However, it would be challenging to teach our models when to introduce these abstractions and how to use them, as there are not many subgraphs in our dataset to begin with. Nonetheless, generating graphs and subgraphs in a top-down or bottom-up manner would not conflict with our tree search or representation and would be an interesting direction for future work.
>
> > Q2: Rather than 1 node at a time, would it be possible for each iteration to generate a set of nodes all residing at the same topological sort depth? [...]
>
> That is an interesting idea. In terms of the gradual shift from purely auto-regressive graph generation toward iterative node generation, we find that this approach would fall between MultiMat (mixed) and MultiMat (graph). If we interpolate from our results, we would expect its performance to lie somewhere between these two variants as well. We believe this would be worth exploring in future work.
>
> > Q3: Were the error patterns of Section 4.3 identified once (or a few times) through manual intervention, then hardcoded into the pipeline? [...]
>
> The error patterns were identified through qualitative analysis and subsequently integrated into the inference algorithm. We appreciate the reviewer's suggestion regarding online feedback support. This could potentially be implemented using multiple agents, with one responsible for generation and another monitoring the process to identify issues in real-time. We believe this represents a promising direction for future work.
>
> > Q4: To my understanding, the iterative generation scheme provides information about the (partial) graph's validity and state visualization/description; however, it does not evaluated or provide feedback on e.g., whether the generated image is likely to lead to the target image under conditional generation. It seems like that would be an interesting extension though; would it be possible to evaluate and/or include any such information to shape the trajectory of the generation procedure? Do there exist any heuristics to judge the suitability of a partial solution for this domain?
>
> While our approach does not provide explicit feedback, we believe that the visualizations make it easier for the model to learn this implicitly. However, we also think this could be augmented by making the feedback explicit through an extension of our tree search algorithm. If we use one of our perceptual similarity metrics from our evaluation as a reward signal, we could extend our tree search algorithm to, for example, a Monte Carlo tree search. By repeatedly running simulations, we would refine the generated graph to better match the input conditions.

---

### Author Response · Authors · 2025-11-18
**General Response**

We thank all reviewers for their time and comments. We have updated our manuscript (new text is highlighted in orange), added examples, conducted a human evaluation, and addressed reviewer-specific comments in the corresponding threads below. Here, we provide answers to concerns shared among multiple reviewers. We hope we have adequately addressed all concerns and are happy to provide additional details if required.

> more qualitative comparisons, human evaluation (Reviewers Zyod, d96n)

We agree that human evaluation and additional examples would strengthen our results. We conducted a human evaluation study with several procedural material experts and have included the findings in our revision. The results confirm the trends observed in our automatic evaluation. We have also added more qualitative comparisons (including more failure cases) in the appendix, which further support these trends. In the appendix you can also find visualized graphs of generated materials, and unconditionally generated materials. We are currently working on the videos of 3D PBR renders and will make them available in the supplementary materials shortly.

> more formal definition of the training objective and sampling strategy (Reviewers nJcx, d96n)

While the training objective and sampling strategy follow standard procedures in LLMs and are often omitted from papers, we agree that explicit documentation improves clarity. Our training uses a standard cross-entropy loss:

$$\mathcal{L} = -\sum_{t=1}^{T} \log p(y_t | y_{<t}, x)$$

where $y_t$ is the expected token at position $t$, $y_{<t}$ represents previous tokens, and $x$ denotes the input conditions. For MultiMat, $y_{<t}$ is reset and $x$ is updated after each node. Token generation proceeds autoregressively, with each token conditioned on previously generated tokens and input conditions:

$$y_t \sim p(y_t | y_{<t}, x; \theta)$$

where $\theta$ represents the model parameters. As is standard, we do not employ negative sampling. The combination of image and text embeddings depends on the underlying base models. The Qwen models we use implement a late fusion approach: a vision encoder extracts visual features, which are then projected into the language model's embedding space for joint processing with text. We have incorporated these technical details into our paper to improve clarity.

> efficiency trade-off; longer training times of MultiMat compared to VLMaterial (Reviewers Zyod, tCBc, d96n)

We acknowledge this concern but believe its practical impact is limited. Regardless of the node graph system, the available training data for procedural materials is very small. Consequently, training times remain within acceptable bounds in absolute terms, despite the relative differences between methods. That said, we acknowledge the importance of addressing this efficiency gap to increase the method's applicability to other domains. Current LLM training frameworks are optimized for purely autoregressive training, as used in VLMaterial. Improving these frameworks for MultiMat's novel training setup is something we want to address in future work. We anticipate considerable speedups through improved caching techniques, e.g. for vision encoder outputs. We have added this clarification to our revision.

> Evaluation on one environment (Substance Designer), with generalization to other systems unclear (Reviewers d96n, Zyod, tCBc)

We acknowledge the concerns about generalizability. However, by focusing on a single environment we follow established practices in this domain [1,2,3], as integrating multiple environments requires considerable engineering effort that extends beyond the scope of a research contribution. By basing our experiments on Substance Designer, we utilize one of the most complex graph-based tools available. Since our method handles this challenging environment effectively, we believe that it would transfer successfully to other graph-based environments.

[1] [VLMaterial: Procedural Material Generation with Large Vision-Language Models](https://iclr.cc/virtual/2025/poster/27856)
[2] [Generating Procedural Materials from Text or Image Prompts](https://dl.acm.org/doi/10.1145/3588432.3591520)
[3] [MatFormer: A Generative Model for Procedural Materials](https://dl.acm.org/doi/abs/10.1145/3528223.3530173)

---

> ### Author Response · Authors · 2025-11-25
> **Update to General Response**
>
> We are pleased to inform the reviewers that we have updated our submission to include a video showcasing 3D PBR renders of procedural materials generated with MultiMat in the supplementary materials. The video demonstrates realistic light interaction and detailed surface properties of these materials. Additionally, we have expanded our manuscript to incorporate results from additional human evaluators in our human evaluation. These new results remain consistent with our previous findings and further validate the trends observed in our automatic evaluation.

---

> > ### Author Response · Authors · 2025-11-26
> > **Update to General Response**
> >
> > We have expanded our analysis section with a deeper investigation of our error repair mechanism. Our findings reveal that only a small fraction of nodes generated by the MultiMat models require correction: approximately 1% of nodes contain hallucinated parameters, and fewer than 6.5% require type conversion. In contrast, VLMaterial exhibits nearly double these error rates for both repair mechanisms. This difference demonstrates that VLMaterial requires considerably more post-processing than our models, supporting our claim that our approach achieves superior understanding of procedural graph structures. Importantly, since corrections are not fed back to the models during generation, these results reflect their intrinsic generation capabilities. We believe these findings strengthen our paper's contributions.

---

### Author Response · Authors · 2025-12-02
**Message to the ACs**

In light of recent events and the valuable discussions we have had with the reviewers, we would like to provide an overview of the rebuttal discussion. Overall, all reviewers praised the core idea of our work. In response to our initial submission, some reviewers raised clarifying questions about our methods. Hence, we have added additional details where necessary throughout the paper. Changes are highlighted in orange for convenience.

Additionally, some reviewers requested more qualitative examples and a human evaluation. We have therefore conducted a human evaluation that confirms the trends observed in our automatic evaluation, and the qualitative examples we added support this direction as well. We have also included a video of 3D renderings in the supplementary materials to illustrate the properties of the generated PBR materials.

Finally, some reviewers expressed interest in additional ablation studies of certain subcomponents of our method. Following this suggestion, we have clarified that some of these ablations were already included and expanded our analysis section with more comprehensive investigations of the remaining components, further validating our main findings.

Reviewers nJcx and tCBc were supportive of our work from the beginning. Reviewer Zyod appreciated our changes and indicated they would increase their score. While we have addressed all of Reviewer d96n's concerns, we unfortunately did not receive a response from them in time due to current circumstances.

---

### Meta-Review · Area_Chair_Rsph · 2025-12-24

**Summary:**

The paper presents MultiMat, a framework for generating procedural material graphs using MLLMs.  Unlike prior work which generates the material graph directly as textual program code, MultiMat models the material graph and generate and visually validates each node iteratively, using a tree search.  The paper also contributes a dataset of 6000 materials from the Adobe Substance 3D Assets Repository.

Reviewers are split on this work with two marginal accepts (nJcx,tCBc) and two rejects (reject - Zyod, marginal reject - d96n) with the most negative reviewer indicating willingness to increase their score.  While reviewers had a number of concerns, the AC believes most of them has been addressed by the author response and revised manuscript.  As such, the AC recommends acceptance.

**Reviewer Concerns:**

Reviewers expressed the following concerns:
1. Missing details [nJcx, d96n]
2. Missing ablations / unclear if important ablations are included (e.g. iterative node generation vs single shot) [nJcx, d96n, Zyod]
3. Reliance on automatic metrics, missing human evaluation [Zyod]
4. Lack of qualitative examples [d96n]
5. Potentially missing discussion of some works on multimodal program synthesis [d96n]
6. Work is limited to a single-system (Substance Designer) [d96n, tCBc,Zyod]
7. Training cost and inefficiency [tCBc,Zyod]

The authors revised the manuscript to address points 1-5 including: 1) provide additional details, 2) add clarification of experiments that are intended to compare different design choices -- e.g. VLMaterial (single pass generation of the graph) and the proposed MultiMat (iterative generation), and discussion of importance of tree search and error repair in Section 7, 3) addition of human evaluation (section 6.3), 4) addition of qualitative examples in the appendix, and 5) included additional prior work suggested by reviewer d96n.

The last two concerns (6,7) are discussed as limitations of the work and beyond the scope of this work to resolve.   Overall, the AC believe the revisions sufficiently addressed the concerns from reviewers.

**Reviewer Scores:**

The paper received mixed scores from reviewers with two marginal accept (nJcx,tCBc) and two on the reject side (reject - Zyod, marginal reject - d96n).  Reviewer Zyod indicated some of their concerns was addressed and was willing to update their score despite still having some concerns.  The AC believes this to a signal that they will increase to marginal reject (4) or potentially marginal accept (6).

---

### Decision · Program_Chairs · 2026-01-26

Accept (Poster)